

# Identification of hub genes and molecular mechanisms in infant acute lymphoblastic leukemia with *MLL* gene rearrangement

Hao Zhang[*], Juan Cheng[*], Zijian Li and Yaming Xi

Department of Hematology, The First Hospital of Lanzhou University, Lanzhou, Gansu, China
[*] These authors contributed equally to this work.

## ABSTRACT

Infant acute lymphoblastic leukemia (ALL) with the mixed lineage leukemia (*MLL*) gene rearrangement (*MLL*-R) is considered a distinct leukemia from childhood or non-*MLL*-R infant ALL. To detect key genes and elucidate the molecular mechanisms of *MLL*-R infant ALL, microarray expression data were downloaded from the Gene Expression Omnibus (GEO) database, and differentially expressed genes (DEGs) between *MLL*-R and non-*MLL*-R infant ALL were identified. Gene ontology (GO) and Kyoto Encyclopedia of Genes and Genomes (KEGG) pathway enrichment analyses were carried out. Then, we constructed a protein-protein interaction (PPI) network and identified the hub genes. Finally, drug-gene interactions were mined. A total of 139 cases of *MLL*-R infant ALL including 77 (55.4%) fusions with *AF4*, 38 (27.3%) with *ENL*, 14 (10.1%) with *AF9*, and 10 (7.2%) other gene fusions were characterized. A total of 236 up-regulated and 84 down-regulated DEGs were identified. The up-regulated DEGs were mainly involved in homophilic cell adhesion, negative regulation of apoptotic process and cellular response to drug GO terms, while down-regulated DEGs were mainly enriched in extracellular matrix organization, protein kinase C signaling and neuron projection extension GO terms. The up-regulated DEGs were enriched in seven KEGG pathways, mainly involving transcriptional regulation and signaling pathways, and down-regulated DEGs were involved in three main KEGG pathways including Alzheimer's disease, TGF-beta signaling pathway, and hematopoietic cell lineage. The PPI network included 297 nodes and 410 edges, with *MYC*, *ALB*, *CD44*, *PTPRC* and *TNF* identified as hub genes. Twenty-three drug-gene interactions including four up-regulated hub genes and 24 drugs were constructed by Drug Gene Interaction database (DGIdb). In conclusion, *MYC*, *ALB*, *CD44*, *PTPRC* and *TNF* may be potential bio-markers for the diagnosis and therapy of *MLL*-R infant ALL.

## INTRODUCTION

Infant acute lymphoblastic leukemia (ALL) refers to ALL arising in infants prior to 12 months of age. Infant ALL is less common but more aggressive than pediatric ALL, generally with a poorer outcome (*Brown, Pieters & Biondi, 2019*; *Pieters et al., 2007*). Despite advances in the treatment of pediatric ALL, >50% of patients with infant ALL

Corresponding author
Yaming Xi, ldyyxyk@gmail.com

relapse within five years of diagnosis, and the four-year event-free survival is <50% (*Hilden et al., 2006*; *Nagayama et al., 2006*). Approximately 80% of infant ALL cases are characterized genetically by rearrangements in the mixed lineage leukemia gene (*MLL*, also known as *KMT2A*, located on chromosome 11q23) (*Krivtsov & Armstrong, 2007*). These rearrangements occur in nearly 100% of infants with congenital leukemia and approximately 5% of pediatric ALL patients, with predicted inferior outcomes (*Van der Linden et al., 2009*). Previous studies have shown that infant ALL with *MLL* rearrangements (*MLL*-R) were clinically distinct, and were characterized by high white blood cell counts, hepatosplenomegaly, and central nervous system and skin involvement (*Hilden et al., 2006*). The 4-5-year event-free survival for *MLL*-R infant ALL patients was 29.1%–43.2%, compared with 56.9%–95.5% for patients without *MLL* rearrangements (non-*MLL*-R) (*Guest & Stam, 2017*). Standard strategy for *MLL*-R infant ALL is intensive chemotherapy with or without hematopoietic stem cell transplantation (HSCT). Although allogeneic HSCT may contribute to improve the inferior prognosis of *MLL*-R infant ALL, it was restricted to some patients because of early relapse before HSCT (*Tomizawa et al., 2007*). Early (within 4 months of the first induction course) use of allogeneic HSCT could help to reduce the rate of early relapse, however, could not ovecome the high risk factors including younger than 90 days at diagnosis, central nervous system involvment and poor prednisone response (*Koh et al., 2015*). Recent studies evaluated the effect of several innovative approaches such as *FLT3* inhibitors, epigenetic agents and immunotherapy in *MLL*-R infant ALL, the results were still far from satisfactory (*Brown, Pieters & Biondi, 2019*). Thus novel treatment targets and molecular-targeted strategies are required to improve outcomes of infant ALL patients with *MLL*-R.

More than 90 different *MLL* partner genes have been identified to date. Frequent *MLL* rearrangements in infant ALL include fusions with *AF4* (49%), *ENL* (22%), *AF9* (16%), and *AF10* (6%) (*Meyer et al., 2018*). All types of rearrangements in *MLL* were independently associated with an unfavorable prognosis (*Pieters et al., 2007*). Several studies found that *MLL* rearrangements occurred in utero, resulting in rapid progression to leukemia (*Brown, Pieters & Biondi, 2019*; *Ford et al., 1993*; *Gale et al., 1997*). Notably, this phenomenon showed a high concordance rate between identical twins (*Greaves et al., 2003*; *Guest & Stam, 2017*). These findings revealed that *MLL* rearrangements may initiate leukemogenesis for *MLL*-R infant ALL. However, *MLL-AF4* expression alone was not sufficient to induce leukemia in human embryonic stem cell-derived hematopoietic cells, and additional genetic candidates were required (*Stam, 2013*). These results suggested that the mechanisms responsible for *MLL*-R infant ALL are distinct from those acting during leukemogenesis in childhood and non-*MLL*-R infant ALL.

We investigated the molecular mechanisms of *MLL*-R infant ALL by determining differentially expressed genes (DEGs) between *MLL*-R and non-*MLL*-R infant ALL, using available microarray datasets, followed by bio-functional enrichment of identified DEGs. These results will provide new insights into the molecular mechanisms behind ALL in infants with *MLL* rearrangements, and help to identify new diagnostic bio-markers and candidate therapeutic targets.

## MATERIALS & METHODS

### Microarray data collection

Microarray expression data in the Gene Expression Omnibus (GEO) database (http://www.ncbi.nlm.nih.gov/geo) (*Barrett et al., 2013*), ArrayExpress database (https://www.ebi.ac.uk/arrayexpress/) and The Cancer Genome Atlas (TCGA) database (https://cancergenome.nih.gov/), were searched using the keywords "acute lymphoblastic leukemia", and data containing expression profiles of *MLL*-R compared with non-*MLL*-R infant ALL cases were selected manually. Raw CEL files were downloaded for further analysis.

### Identification of DEGs

Gene expression profile data were preprocessed using the robust multi-array average (RMA) algorithm by Affy package in R (version 1.58.0) (*Gautier et al., 2004*), including background adjustment, normalization, and summarization. According to annotation files, the mean value was computed for several probes matched to a specific gene, and used as the expression value of that gene. DEGs between *MLL*-R and non-*MLL*-R infant ALL cases were identified using the Bayesian method by Limma package in R (version 3.36.5) (*Ritchie et al., 2015*). |log2 fold change (FC)|>1 (log2FC >1 defined as upregulated genes, log2FC <−1 defined as downregulated genes) and a *P* value <0.01 were considered as threshold points.

### Gene functional enrichment analysis

Gene ontology (GO) functional annotation analyses including biological processes (BP), cellular components (CC), molecular function (MF) terms and Kyoto Encyclopedia of Genes and Genomes (KEGG) pathway analysis were performed using the Database for Annotation, Visualization and Integrated Discovery (DAVID) v6.8 (*Huang da, Sherman & Lempicki, 2009*), with a default cut-off criterion of count ≥2 and *P* value <0.1.

### Protein-protein interaction network construction and analysis

A protein-protein interaction (PPI) network of DEGs was constructed using the STRING (version 10.5, http://www.string-db.org/) database (*Von Mering et al., 2003*), with minimum required interaction score >0.4 (median confidence). The PPI network was visualized using Cytoscape (version 3.7.1, http://www.cytoscape.org/) (*Shannon et al., 2003*). Bio-functional modules in the PPI network were explored using a plug-in MCODE (version 1.4.2, http://apps.cytoscape.org/apps/MCODE) in Cytoscape with Node Score Cutoff of 0.2 and K-Core of 2. Hub genes were screened using the plug-in CytoHubba (version 2.1.6, http://apps.cytoscape.org/apps/cytohubba) in Cytoscape with methods including maximal clique centrality, degree, and betweenness.

### Drug-gene interactions analyses

Drug-gene interactions were searched from the Drug-Gene Interaction database (DGIdb, v3.0.2, http://www.dgidb.org/) (*Cotto et al., 2018*), which mines known or predicted interactions from existing databases and literature, using the list of hub genes. The preset filter was setted to antineoplastic which was defined by the inclusion of anti-neoplastic
drug-gene interaction source databases (e.g., My Cancer Genome, PharmGKB, DrugBank), while advanced filters were setted to 20 source databases, 41 gene categories and 51 interaction types. The interactions were visualized using Cytoscape.

## RESULTS

### Microarray datasets and patient characteristics

Based on searches in the GEO, ArrayExpress and TCGA databases, two microarray datasets, GSE68720 and GSE19475, that met the criteria mentioned in methods section, were selected for analysis. Both datasets were generated using the GPL570 Affymetrix Human Genome U133 Plus 2.0 Array platform. There were 80 *MLL*-R and 17 non-*MLL*-R infant ALL samples in the GSE68720 dataset, and 59 and 14, respectively, in the GSE19475 dataset. The age range of the 139 *MLL*-R infant ALL cases (65 male, 73 female and one unknown) was 0–363 days, and the age range of the 31 non-*MLL*-R infant ALL cases (22 male, nine female) was 0–365 days. *MLL*-R infant ALL displayed 77 (55.4%) *AF4*, 38 (27.3%) *ENL*, 14 (10.1%) *AF9*, and 10 (7.2%) other gene fusions.

### Identification of DEGs

To probe significant alterations of gene expression profile related to *MLL*-R, differential expression analysis of genes was performed using Limma package. Based on the cut-off criteria, a total of 320 DEGs, including 236 up-regulated and 84 down-regulated genes, were identified between *MLL*-R and non-*MLL*-R infant ALL samples. The top five up-regulated DEGs were *LAMP5*, *PROM1*, *CCNA1*, *MEIS1*, and *KCNK12*, and the top five down-regulated DEGs were *MME*, *CMTM8*, *ELK3*, *NRN1*, and *PLEKHG4B* (Fig. 1).

### GO functional annotation and KEGG pathway analyses

To identify the biologic functions of DEGs, GO functional annotation and KEGG pathway analyses were carried out. The results showed that up-regulated DEGs were enriched in 62 BP, 21 CC, and 22 MF terms, and down-regulated DEGs were enriched in 34 BP, 22 CC, and 17 MF terms. The top five significant terms are shown in Figs. 2, 3 and 4. The up-regulated DEGs were also enriched in seven KEGG pathways, mainly involving transcriptional regulation and signaling pathways, and down-regulated DEGs were enriched in three KEGG pathways including Alzheimer's disease, TGF-beta signaling pathway, and hematopoietic cell lineage (Table 1).

### PPI network construction and analysis

The STRING online database and Cytoscape were used to detected interactions between DEGs-encoded proteins by constructing PPI network. The PPI network of DEGs included 297 nodes and 410 edges (Fig. 5). Five bio-functional modules were screened out, based on the cut-off criteria, and the maximal MCODE score of module, which consisted of 15 nodes and 75 edges, was 10.714. This module contained 11 up-regulated (e.g., *MYC*, *CD44*, and *PTPRC*, etc.) and four down-regulated (e.g., *TNF*, *MME*, and *RAG1*, etc.) DEGs (Fig. 6). The top 10 hub genes identified by Cytoscape, using maximal clique centrality, degree and betweenness, are listed in Table 2. *MYC*, *ALB*, *CD44*, *PTPRC* and *TNF* were identified as hub genes based on all three methods.

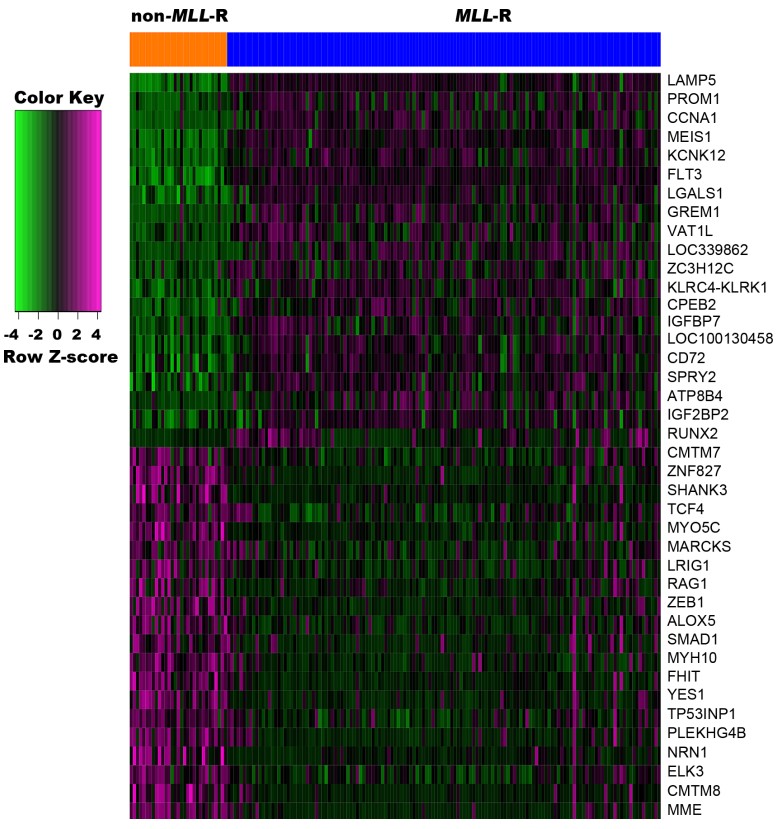

**Figure 1  Heat map of the top 20 up-regulated and down-regulated DEGs.** DEGs were identified between *MLL*-R (*n* = 139) and non-*MLL*-R (*n* = 31) infant ALL samples by the cutoff | log2 fold change (FC) | > 1 and *P* value <0.01. Each row represents a single gene, each column represents a sample. The gradual color change from green to magenta represents the gene expression values change from low to high. DEG, differentially expressed genes; *MLL*-R, Mixed Lineage Leukemia rearrangement.

## Drug-gene interactions analyses

To explore potentially drugs for patients with *MLL*-R, the network of drug-gene interactions were constructed by DGIdb with 29, 783 drug-gene interactions consist of 41,100 genes and 9,495 drugs. As a result, 23 drug-gene interactions including 4 up-regulated hub genes (*MYC*, *ALB*, *CD44*, *PTPRC*) and 24 drugs were identified, as shown in Fig. 7.

## DISCUSSION

MLL, encoded by the histone-lysine N-methyltransferase 2A (*KMT2A*), is a transcriptional coactivator that binds with other proteins in complex and methylates histone H3 lysine 4 which acts as an instructive mark for transcription initiation (*Mohan et al., 2010*). Rearrangements of *MLL* results in the fusion of its N-terminus with the C-terminus of a partner gene, leading to transcription dysregulation (*Armstrong, Golub & Korsmeyer, 2003*). Previous research identified more than 90 fusion genes of *MLL*, including the frequent *MLL* rearrangement in infant ALL t(4;11)(q21;q23), which results in the *MLL-AF4* fusion (*Meyer et al., 2018*; *Meyer et al., 2013*). In the present study, three partner genes accounted

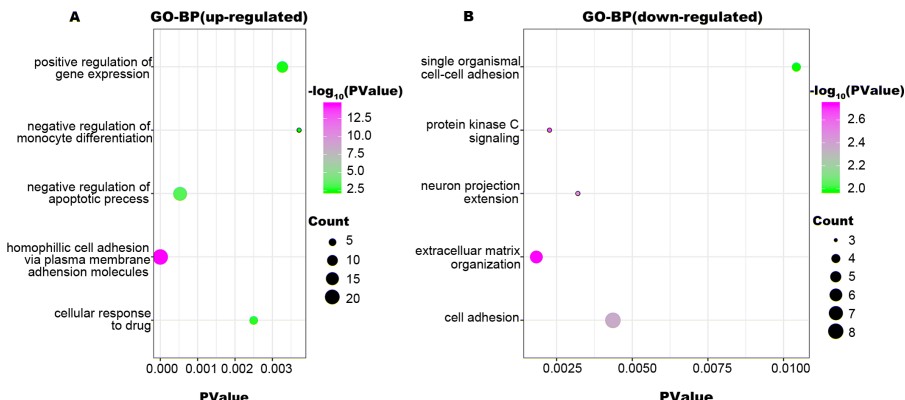

**Figure 2** **GO-BP function annotation the DEGs.** The up-regulated DEGs were enriched in 62 BP, terms, and down-regulated DEGs were enriched in 34 BP terms with a cut-off criterion of count ≥2 and *P* value <0.1. The gradual color change from green to magenta represents the −log10(P Value) change from low to high, the size of point represents the the count of genes. (A) The top five significantly enriched GO-BP terms for up-regulated DEGs. (B) The top five significantly enriched GO-BP terms for down-regulated DEGs. DEG: differentially expressed genes; GO, gene ontology; BP, biological process.

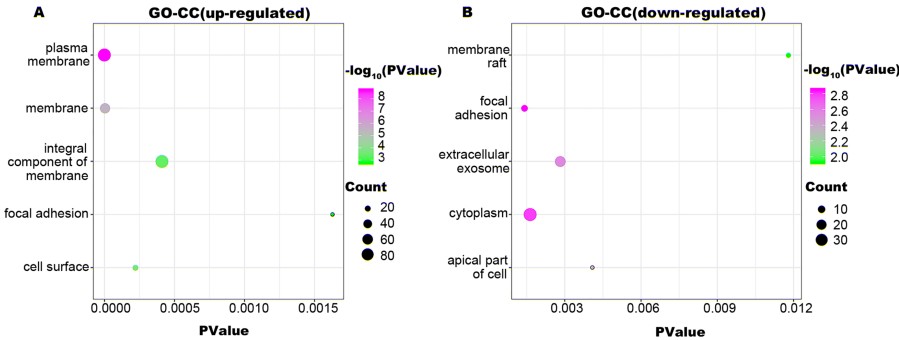

**Figure 3** **GO-CC function annotation the DEGs.** The up-regulated DEGs were enriched in 21 CC terms, and down-regulated DEGs were enriched in 22 CC terms with a cut-off criterion of count ≥2 and *P* value <0.1. The gradual color change from green to magenta represents the −log10(P Value) change from low to high, the size of point represents the the count of genes. (A) The top five significantly enriched GO-CC terms for up-regulated DEGs. (B) The top five significantly enriched GO-CC terms for down-regulated DEGs. DEG, differentially expressed genes; GO, gene ontology; CC, cellular component.

for 92.8% of 139 cases: *MLL-AF4* (55.4%), *MLL-ENL* (27.3%), and *MLL-AF9* (10.1%). To confirm that the molecular mechanisms responsible for *MLL*-R infant ALL differs from those of childhood or non-*MLL*-R, we analyzed differences in gene expression profiles between *MLL*-R and non-*MLL*-R infant ALL samples, based on microarray datasets obtained from the GEO database. Two microarray datasets were selected, and a total of 320 DEGs with fold-change >2 were screened out, including 236 up-regulated genes (e.g., *LAMP5*, *PROM1*, *CCNA1*, *MEIS1*, and *KCNK12*) and 84 down-regulated genes (e.g., *PLEKHG4B*, *NRN1*, *ELK3*, *CMTM8*, and *MME*). These results provide preliminary evidence for the distinct mechanisms of *MLL*-R infant ALL. We further investigated the

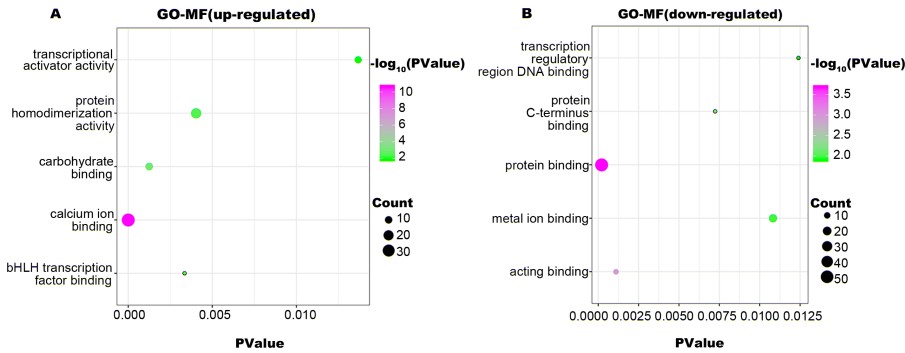

**Figure 4 GO-MF function annotation the DEGs.** The up-regulated DEGs were enriched in 22 MF terms, and down-regulated DEGs were enriched in 17 MF terms with a cut-off criterion of count ≥2 and *P* value <0.1. The gradual color change from green to magenta represents the −log10(PValue) change from low to high, the size of point represents the the count of genes. (A) The top five significantly enriched GO-MF terms for up-regulated DEGs. (B) The top five significantly enriched GO-MF terms for down-regulated DEGs. DEG, differentially expressed genes; GO, gene ontology; MF, molecular function.

**Table 1 KEGG pathways of DEGs.**

| Term | Genes | Type of DEGs | *P* value |
|------|-------|--------------|-----------|
| Transcriptional misregulation in cancer | *MEF2C, PROM1, CEBPA, LMO2, FLT3, RUNX1, HMGA2, RUNX2, MEIS1, MYC, WT1* | up-regulated | 0.000 |
| Acute myeloid leukemia | *CEBPA, FLT3, RUNX1, MYC* | up-regulated | 0.042 |
| Oxytocin signaling pathway | *MEF2C, CAMK2D, GUCY1A3, PPP3CA, NFATC3, CACNA2D4* | up-regulated | 0.056 |
| Proteoglycans in cancer | *CD44, RRAS2, CAMK2D, HBEGF, PTCH1, MYC, SDC2* | up-regulated | 0.058 |
| Renin secretion | *PTGER2, PTGER4, GUCY1A3, PPP3CA* | up-regulated | 0.058 |
| MAPK signaling pathway | *MEF2C, MAP3K5, RRAS2, GNA12, PPP3CA, NFATC3, MYC, CACNA2D4* | up-regulated | 0.059 |
| Pathways in cancer | *CEBPA, PTGER2, PTGER4, FLT3, GNA12, SMAD3, PTCH1, CDK6, RUNX1, MYC* | up-regulated | 0.092 |
| MicroRNAs in cancer | *SPRY2, CD44, VIM, ZEB2, CDK6, HMGA2, MYC, DDIT4* | up-regulated | 0.099 |
| Alzheimer's disease | *APP, TNF, MME, ITPR3* | down-regulated | 0.053 |
| TGF-beta signaling pathway | *TNF, ID3, SMAD1* | down-regulated | 0.068 |
| Hematopoietic cell lineage | *TNF, MS4A1, MME* | down-regulated | 0.072 |

**Notes.**
KEGG, Kyoto Encyclopedia of Genes and Genomes; DEG, differentially expressed genes.

functions of these DEGs and their possible roles in *MLL*-R infant ALL using DAVID functional annotations. In GO-BP, the top three significant terms were homophilic cell adhesion, negative regulation of apoptosis and cellular responses to drugs. Homophilic cell adhesion, mediated by the up-regulation of protocadherin (*PCDH*) γ subfamily of genes (e.g., *PCDHGA*, *PCDHGB*, *PCDHGC*) may contribute to cell migration and invasion. Previous studies have confirmed that *PCDH* genes were involved in tumorigenesis and metastasis of gastric cancer, follicular lymphoma and non-small cell lung cancer (*Mukai et al., 2017*; *Zhang et al., 2016*; *Zhou et al., 2017*), although other studies have found that

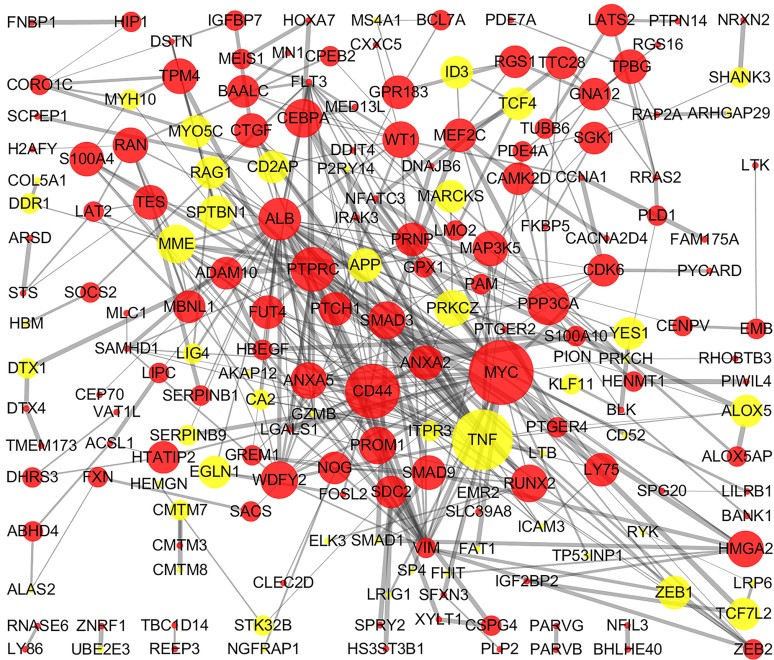

**Figure 5** **PPI network of DEGs.** The PPI network of DEGs included 297 nodes and 410 edges with minimum required interaction score >0.4 (median confidence). The size of edges change from small to large represents the combined score of nodes change from low to high, the size of nodes represents the count. Red nodes represent up-regulated genes, yellow nodes represent down-regulated genes. PPI, protein–protein interaction; DEG, differentially expressed genes.

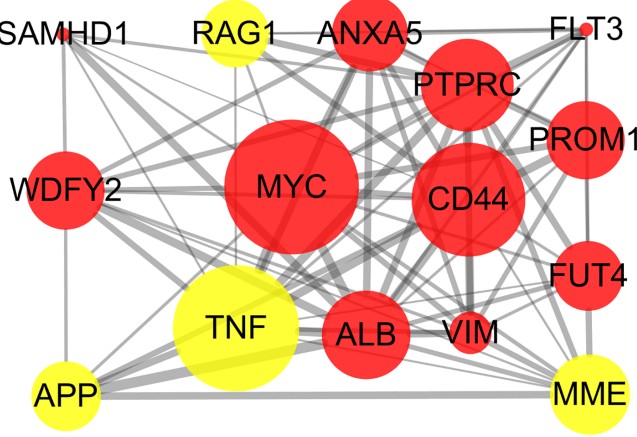

**Figure 6** **Bio-functional module of DEGs.** The bio-functional module with the maximal MCODE score contained 11 up-regulated and four down-regulated DEGs. The size of edges change from small to large represents the combined score of nodes change from low to high, the size of nodes represents the count. Red nodes represent up-regulated genes, yellow nodes represent down-regulated genes. DEG, differentially expressed genes; MCODE, Molecular COmplex Detection.

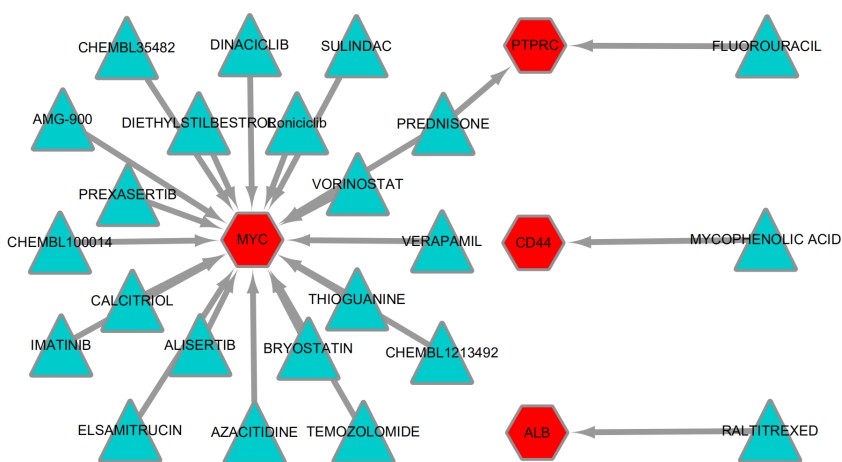

**Figure 7** **Drug-gene interactions.** Drug-gene interactions. Twenty-three drug-gene interactions including 4 up-regulated hub genes and 24 drugs were identified by DGIdb. Red nodes represent up-regulated hub genes and green nodes represent the drugs. DGIdb: Drug-Gene Interaction database.

**Table 2** **Top 10 hub genes that evaluated by the methods including MCC (Maximal Clique Centrality), Degree and Betweenness.**

| Gene | MCC score | Gene | Degree score | Gene | Betweenness score |
|------|-----------|------|--------------|------|-------------------|
| MYC | 22,218 | MYC | 43 | ALB | 8,542.027612 |
| CD44 | 22,075 | ALB | 42 | MYC | 8,398.777703 |
| ALB | 21,713 | TNF | 33 | TNF | 4,702.068242 |
| PTPRC | 21,310 | CD44 | 26 | CD44 | 4,164.511092 |
| TNF | 19,364 | PTPRC | 22 | PPP3CA | 2,262.216619 |
| VIM | 16,980 | VIM | 20 | SDC2 | 2,006 |
| WDFY2 | 11,570 | SMAD3 | 18 | FLT3 | 1,790.958256 |
| MME | 10,849 | FLT3 | 17 | MAP3K5 | 1,762.347815 |
| ANXA5 | 8,334 | ANXA5 | 15 | SMAD3 | 1,738.446661 |
| FUT4 | 7,200 | MME | 14 | PTPRC | 1,709.765874 |

*PCDH* genes were implicated in tumor suppression (*Chen et al., 2017*; *Weng et al., 2018*). However, the role of *PCDH* genes in *MLL*-R ALL is currently unclear. SOCS2, one of the up-regulated DEGs enriched in negative regulation of apoptosis, is demonstrated to be a feedback inhibitor of JAK-STAT signaling pathways. High levels of SOCS2 was identified as an adverse prognostic characteristic of acute myeloid leukemia (AML) and ALL, such as those with *MLL* rearrangement or *BCR/ABL* fusions (*Hansen et al., 2013*; *Vitali et al., 2015*). *MEF2C, RAP2A, PDE4A, PPP3CA, PRNP,* and *MYC* were enriched in the GO-BP term cellular responses to drugs which may be involved in chemoresistance. As a transcription factor related to normal hematopoiesis, myocyte enhancer factor 2C (MEF2C) was involved in a number of transcriptional complexes (*Cante-Barrett, Pieters & Meijerink, 2014*). *MEF2C* was significantly up-regulated in acute leukemia with *MLL*-R, while known as an adverse prognostic marker in AML (*Laszlo et al., 2015*). Recent research

suggested that *MEF2C* was related to AML chemoresistance caused by the phosphorylation of MEF2C S222 (*Brown et al., 2018*). *PDE4A*, encodes one of phosphodiesterases (PDEs), is abundantly expressed in leukemia cell lines. Inhibitors of *PDE4* augmented apoptosis induced by glucocorticoid and overcame the resistance to glucocorticoid in leukemia cells (*Dong et al., 2010*; *Ogawa et al., 2002*).

We further uncovered molecular mechanistic insights of *MLL*-R ALL by KEGG pathway analysis, using the DAVID tool. The most significant pathway affected was transcriptional misregulation in cancer, which was enriched with up-regulated DEGs, including *MEF2C*, *PROM1*, *CEBPA*, *LMO2*, *FLT3*, *RUNX1*, *HMGA2*, *RUNX2*, *MEIS1*, *MYC*, and *WT1*. Some of these genes have been defined as *MLL* targets genes. MEIS1, which is known as a key regulator in transcriptional regulation, cellular differentiation, and cell-cycle control, exhibited significant increases in both MLL fusion protein binding and mRNA expression on *MLL-ENL* activation in *MLL-ENL* leukemia cases and in an inducible cellular model (*Wang et al., 2011*). Further research also indicated that MEIS1 was essential for the development of *MLL* leukemia, by promoting cell differentiation resistance, and it was also confirmed to be involved in chemotherapy resistance (*Rosales-Avina et al., 2011*). As a downstream gene, *HMGA2* was positively regulated by MLL fusion proteins in infant *MLL-AF4* ALL leukemic cells (*Wu et al., 2015*). *PROM1* is a commonly used stem cell and cancer stem cell marker, and *MLL-AF4* was shown to promote *PROM1* transcription, which is required for *MLL-AF4*-driven leukemia cell growth (*Mak, Nixon & Moffat, 2012*). FLT3 is a class III receptor tyrosine kinase that plays an important role in hematopoietic stem cell development, high levels of *FLT3* are a common cooperating event in *MLL-AF4* ALL (*Bueno et al., 2013*). The presence of activating *FLT3* mutations in *MLL* is in keeping with a multistep pathway of leukemogenesis, suggesting that *FLT3* mutations may act as a second hit to lead to leukemogenesis in *MLL*-R infants (*Kang et al., 2012*). *LMO2* plays an important role in hematopoiesis and leukemogenesis, Begay-Muller's research indicated that AF6, a recurrent fusion partner of *MLL*, bound to *LMO2* and may be involved in mixed lineage leukemia (*Begay-Muller, Ansieau & Leutz, 2002*). *RUNX1* is known to be a putative target gene of *MLL* fusions, and was highly expressed in *MLL-AF4* ALL cases, when compared with normal bone marrow cells (*Guenther et al., 2008*; *Krivtsov et al., 2008*). The *BET* inhibitor I-BET151, arrested the growth of *MLL-AF4* infant leukemic cells *in vitro* through gene deregulation, including *RUNX1* (*Bardini et al., 2018*).

We examined interrelationships among the DEGs by constructing a PPI network using the STRING database and Cytoscape, and identified *MYC*, *ALB*, *CD44*, *PTPRC* and *TNF* as hub genes. *MYC* has been shown to be a direct target of *MLL-AF9* and is differentially expressed between neonatal and adult cells expressing *MLL-AF9* (*Zuber et al., 2011*). Notably, expression of the neonatal, but not the adult *MLL-AF9* signature, was also enriched in a core *MYC* network, suggesting that neonatal cells are inherently more prone to *MLL-AF9*-mediated immortalization than adult cells (*Horton et al., 2013*). Furthermore, MYC was essential for *MLL-ENL* to promote differentiation arrest of myelomonocytic precursor cells, and constitutive MYC expression cooperated with *MLL-ENL* to transform cells, with irreversible maturation arrest (*Schreiner et al., 2001*). *CD44* is a type I transmembrane glycoprotein and a leukocyte marker expressed on hematopoietic cells, various epithelial cell

types, fibroblasts, and endothelial cells. The current findings revealed that CD44 is a surface marker of cancer stem cells (*Yan, Zuo & Wei, 2015*). *MLL*-R ALL had a unique genetic profile clearly distinguishable from those of other types of leukemia, with very high *CD44* levels (*Tsutsumi et al., 2003*). Furthermore, *CD44* expression represented the early steps of lymphoid development, high levels of *CD44* were associated with maturation arrest at an early lymphoid progenitor stage of development, while levels decreased with maturation (*Armstrong et al., 2002*). *PTPRC* gene encodes for the protein tyrosine phosphatase *CD45*, which acts as a haematopoietic *JAK* phosphatase required for lymphocyte activation and development (*Irie-Sasaki et al., 2001*; *Trowbridge & Thomas, 1994*). High levels of *PTPRC* (*CD45*) were associated with poor prednisone response followed by an inferior prognosis in B-cell-precursor ALL and T-cell ALL with *MLL-AF4* (*Cario et al., 2014*). However, the roles of ALB up-regulation and *TNF* down-regulation in *MLL*-R ALL remain unclear. Besides, it should be noted that *NG2* was widely known as a marker and therapeutic target for *MLL*-R leukemia (*Behm et al., 1996*; *Hilden et al., 1997*; *Lopez-Millan et al., 2019*; *Smith et al., 1996*; *Wuchter et al., 2000*), but it was not be identified as hub gene in this study may be caused by the special threshold used for the computational analyses.

In addition, 24 antineoplastic drugs were forecasted based on up-regulated hub genes. Some of these drugs have been used in the treatment of acute leukemia (*Chijimatsu et al., 2017*; *Mosse et al., 2019*; *Schultz et al., 2014*; *Teuffel et al., 2011*). The rest of them could be potential treatment options for infant *MLL*-R ALL.

Nevertheless, it should be noted that the results of this research were obtained by bioinformatics analysis, further experimental and clinical validation should be the focus of our future research.

## CONCLUSIONS

In this study, bioinformatics analyses were performed to detected novel candidate biomarkers and uncover possible molecular mechanisms of *MLL*-R infant ALL. The results indicated that the hub genes *MYC*, *ALB*, *CD44*, *PTPRC* and *TNF*, together with some biological events including negative regulation of apoptotic processes, monocyte differentiation, homophilic cell adhesion, and transcriptional misregulation, may contributed to leukemogenesis, migration, and invasion in *MLL*-R infant ALL. And several drugs were predicted based on the hub genes. All these findings provide novel biomarkers and potential therapeutic approaches for *MLL*-R infant ALL.

## ACKNOWLEDGEMENTS

The authors would like to acknowledge the Gene Expression Omnibus database for providing raw data in this study.

### Funding

The authors received no funding for this work.

## Competing Interests

The authors declare there are no competing interests.

## Author Contributions

- Hao Zhang conceived and designed the experiments, performed the experiments, contributed reagents/materials/analysis tools, prepared figures and/or tables, authored or reviewed drafts of the paper, approved the final draft.
- Juan Cheng performed the experiments.
- Zijian Li analyzed the data, prepared figures and/or tables.
- Yaming Xi authored or reviewed drafts of the paper, approved the final draft.

## Data Availability

The raw data is available at NCBI GEO: GSE68720 and GSE19475.

## Supplemental Information

Supplemental information for this article can be found online at http://dx.doi.org/10.7717/peerj.7628#supplemental-information.

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
