# Peer review of "Identification of hub genes and molecular mechanisms in infant acute lymphoblastic leukemia with MLL gene rearrangement"

_PeerJ, doi:10.7717/peerj.7628_

## Round 0.1 · original submission · Minor Revisions

Dear Dr. Zhang,

Thank you for submitting your manuscript entitled, " Identification of hub genes and molecular mechanisms in infant acute lymphoblastic leukemia with MLL gene rearrangement " to PeerJ. All the reviewers had very positive comments about the work submitted for peer review. Despite, the positive comments two reviewers have raised few minor technical and conceptual concerns which unfortunately preclude publication of the current version of the manuscript in PeerJ. The revised manuscript in response to the review critiques will likely be accepted for publications. Please submit a copy of the revised manuscript with "tracked" or highlighted changes, as well as an unmarked or "clean" version.

We appreciate the opportunity to review your work and look forward to the revised version of the manuscript.

Sincerely,

Sarwat Naz, PhD
Academic Editor

·

Basic reporting

In this manuscript by Zhang et al, the authors have used several bioinformatics tools to elucidate the molecular mechanisms of infant ALL with mixed lineage leukemia gene rearrangement (MLL-R). The objective was to identify potential biomarkers for the diagnosis and therapy of this deadly disease which has a low 5-year event free survival rate.

-The language used in this manuscript was unambiguous and easy to understand. The introduction contained enough background and explained why the study was conducted. This manuscript has been well researched as the literature references provided are extensive. There are some grammatical errors in this manuscript which are noted in the General comments section.

-The results were well presented but brief. More detail in terms of why, what and how should be added. The figures were of good resolution, but the figure legends were lacking in detail. Please add more methodology as well as any statistical analyses performed.

-The discussion section which explained the roles of each of the potential targets (from line 195 onwards) in this disease was well written. Please provide more explanation for the DEGs identified via GO analyses (lines 190-193).

-Lines 251-254 in the Conclusions section should be re-worded.

-All appropriate raw data has been made available.

Experimental design

The authors have used very well known bioinformatics tools to ascertain the molecular mechanisms of MLL-R infant ALL, which defers from childhood and non-MLL-R ALL. Several analyses including KEGG and GO were performed, including the addition of drug-gene interactions to identify and in some cases confirm the "hub" genes in this cancer. This study has original aims which falls within the scope of PeerJ.

-The research question was well defined and provides potential avenues for the identification and targeting of this disease.

-The choice of datasets for further analysis was appropriate and well matched in terms of sex and age so as not be confounding factors.

-Please check the p value for DAVID (line 102).

-More detail should be added to the drug-gene interaction methods. The choice of filters used should be better defined.

-Please add information regarding the statistical analyses (if any) performed.

Validity of the findings

This study fills some of the knowledge gap regarding the DEGs involved MLL-R infant ALL and their differences from other non-MLL-R cancers. The data is robust and well controlled. The normalizations and analyses performed have been sufficiently explained.

-The conclusion for this study states in like 254 that the identified genes "contributed to leukemogenesis, migration, and invasion of" but this is an overstatement since this has not been validated in this study. Please change that sentence to "may contribute".

Additional comments

The tense should be kept constant throughout the text. The following lines contain errors which should be rectified:

Line 18: No period between ALL and microarray expression.
Line 26-28: Needs more clarity.
Line 30: Please mention the KEGG pathways in which the DEGs are downregulated.
Line 33: Define DGIdb.
Line 165: What is the consequence of histone H3 lysine 4 methyation?
Line 223: Please change "to identify" to "and identified".
Line 236: Please comment on the role of CD44 as a marker of cancer stem cells.
Line 240: Please re-word this.
Line 242: Please add "the roles of" before ALB up-regulation.

Overall, please comment on the consequences of your discovery. This is a sound study, but may be lacking some validation.

Reviewer 2 ·

Basic reporting

The manuscript is well-written. However, the second sentence of the abstract is incomplete and requires revision.

Experimental design

No comment

Validity of the findings

No comment

Reviewer 3 ·

Basic reporting

The manuscript is well written in professional scientific language, with very few minor grammatical errors. There is sufficient reference to previously published work in the field with a few suggestions below .The figures are well made and have good descriptions and keys to make their understanding better. The results section is concise, well-written and correctly interprets the findings of their study. There are a few minor flaws as described below:

(1)The introduction is well written and covers enough background about the differences and distinction between pediatric ALL and MLL-R infant ALL. While differences in survival rates and prognoses are mentioned, treatment strategies and current approaches to improve prognosis for this disease (such as work conducted by the Ishii group (Leukemia 2007,Leukemis 2007) needs to be referenced).

(2) Line 18 : grammatical error: the line should read......To elucidate the molecular mechanisms of MLL-R infant ALL, microarray expression datasets were downloaded from the GEO database,......

(3) Figure 1: the x-axis below the figure can be removed or the font size needs to be reduced as it is not legible.

Experimental design

The study conducted in this manuscript fits the scope of the journal. The research question is well defined and addressing a major unmet need in the field of MLL-R infant ALL. The methods section is well written, providing enough background knowledge and details to the reader. The sample and control cohorts identified in this study are well-defined with no skewing factors such as gender or age disparities. The only comment here is

(1) Line 125: please specifically define what "criteria" is.

Validity of the findings

All raw data required to effectively replicate this study has been provided. The conclusions made are true to their findings. The KEGG and DAVID annotation and pathway analysis are very well written. There are few suggestions below on improving the discussion:

(1) Lines 178-182 : Please mention speicifically that all downstream analysis from the DEG list including molecular functions and mechanism is in-silico as no in-vitro validation was performed in this study.

(2) NG2 has bee shown to be a marker for MLL-R infant ALL ( Behm et al, 1996, Blood, Hilden et al, 1997, Blood, Smith et al, 1996, Blood and Wuchter et al, 2000, Leukemia). The authors should comment on why this gene either did not come up as significantly de-regulated in their analysis or how their results correlate with the previous literature.

(3) The GO annotation using DAVID revealed pathways associated with general "hallmarks of cancer" and deregulated cell. While the authors do make some mention on how such findings could be specific to MLL-R infant ALL, more detail will be helpful in fully elucidating the significance of this study.

(4) Please add the appropriate references to line 245.

Additional comments

No comment.

---

## Round 0.2 · accepted · Accept

Dear Dr. Zhang,

I am writing to inform you that your revised manuscript - Identification of hub genes and molecular mechanisms in infant acute lymphoblastic leukemia with MLL gene rearrangement - has received positive reviews from the reviewers. Two reviewers reassessed the revised manuscript and find the current version suitable for publication in PeerJ.

Congratulations!

·

Basic reporting

The authors have made the necessary suggested changes.

Experimental design

No comment.

Validity of the findings

The conclusions have been modified according to suggestions.

Additional comments

All reviewer comments have been addressed. I look forward to validation of these genes in future manuscripts.

Reviewer 3 ·

Basic reporting

All comments have been adequately addressed.

Experimental design

No comment

Validity of the findings

No Comment

Additional comments

No Comment